materials science/nanotechnology

nickel oxide, Co dopant, highly porous nanorods, oxygen evolution reaction, synergistic effect

**Authors for correspondence:**
Nguyen Duc Cuong
e-mail: nguyenduccuong@hueuni.edu.vn
Phong D. Tran
e-mail: tran-dinh.phong@usth.edu.vn

This article has been edited by the Royal Society of Chemistry, including the commissioning, peer review process and editorial aspects up to the point of acceptance.

# Highly porous Co-doped NiO nanorods: facile hydrothermal synthesis and electrocatalytic oxygen evolution properties

Nguyen Duc Cuong[1,2], Tien D. Tran[3], Quyen T. Nguyen[3], Ho Van Minh Hai[1], Tran Thai Hoa[1], Duong Tuan Quang[4], Wantana Klysubun[5] and Phong D. Tran[3]

[1]University of Sciences, Hue University, 77 Nguyen Hue, Hue City, Viet Nam
[2]School of Hospitality and Tourism, Hue University, 22 Lam Hoang, Hue City, Viet Nam
[3]University of Science and Technology of Hanoi, Vietnam Academy of Science and Technology, 18 Hoang Quoc Viet, 100000 Hanoi, Viet Nam
[4]University of Education, Hue University, 34 Le Loi, Viet Nam
[5]Synchrotron Light Research Institute, 111 Moo 6, University Avenue, Muang, Nakhon Ratchasima 30000, Thailand

NDC, 0000-0002-7341-3661

Highly porous 3d transition metal oxide nanostructures are opening up the exciting area of oxygen evolution reaction (OER) catalysts in alkaline medium thanks to their good thermal and chemical stability, excellent physiochemical properties, high specific surface area and abundant nanopores. In this paper, highly porous Co-doped NiO nanorods were successfully synthesized by a simple hydrothermal method. The porous rod-like nanostructures were preserved with the added cobalt dopant ranging from 1 to 5 at% but were broken into aggregated nanoparticles at higher concentrations of additional cobalt. The catalytic activity of Co-doped NiO nanostructures for OER in an alkaline medium was assayed. The 5%Co-NiO sample showed a drastically enhanced activity. This result could originate from the combination of advantageous characteristics of highly porous NiO nanorods such as large surface area and high porosity as well as the important role of Co dopant that could provide more catalytic active sites, leading to an enhanced catalytic activity of the nanocatalyst.

# 1. Introduction

Hydrogen has been considered as a sustainable energy carrier thanks to its high energy density and environmental benignity [1,2]. Water splitting represents a promising route for the large-scale production of $H_2$. It consists of two half-reactions, namely the anodic oxygen evolution reaction (OER) and cathodic hydrogen evolution reaction [3,4]. The OER has slow kinetics, thus requires a high overpotential to sustain an appropriate reaction rate, because of its multiple electron, multiple proton nature being relevant to O–H bond breaking and O=O bond formation [5–7]. Thus, boosting the rate of OER is critical in order to improve the overall efficiency of water electrolysis for $H_2$ generation [8]. To date, noble metal oxide $IrO_2$ and $RuO_2$ catalysts have been considered as the most efficient OER catalysts [9,10]. However, the extensive application of these materials has been impeded by their rareness and high cost. Thus, efforts are being focused to design and develop efficient oxygen evolution catalysts based on elements that are found abundantly in the Earth's crust [11].

3d transition metal oxide-based nanomaterials including iron oxide, nickel oxide, cobalt oxide and manganese oxide have attracted great interest as fascinating OER catalysts in alkaline medium thanks to their good thermal and chemical stability and excellent physiochemical properties [12–14]. $MnO_2$ nanorod arrays showed good activity and stability for OER in alkaline medium because of the enhancement of the $MnO_2$ activity relating to its nanorod array structures [15]. Ni-$Fe_2O_3$ nanoclews exhibited excellent electrocatalytic activity for OER with sustainability and low overpotential due to the atomic-scale synergistic effect arising from Fe and Ni contribution as well as their unique nanostructures promoting highly exposed active catalytic sites [16]. Ultrathin porous $Co_3O_4$ nanoplates with large specific surface area, small crystalline size and high porosity possess great catalytic activity for OER, which may be supported by their structural features that provide more surface-active sites and efficient chemical diffusion [17]. The direct growth of dual active site $NiCo_2O_4$ catalyst on three-dimensional (3D) porous N-doped graphene to form 3D hybrid nanocomposites with some remarkable characteristics such as 3D conductive network and in- and out-of-plane pores can promote charge transport in electrodes, leading to outstanding catalytic efficiency for OER [18]. The results demonstrate that the advantages of nanostructures such as large specific surface area, the increase of edges and defective sites on the surface and high porosity may enhance the intrinsic properties of electrocatalytic materials, subsequently increasing the catalytic OER activity. Thus, the development of a simple method to tailor the composition and control the morphology of metal oxide nanostructures is of considerable interest to explore novel electrocatalytic properties for OER.

Nickel oxide nanostructures, as a promising p-type semiconductor, have been widely used in catalysis [19,20]. In some circumstances, the single nickel oxide showed better catalytic OER performance in comparison with a cobalt, iron or manganese equivalent [21]. However, the nickel oxide-based OER electrocatalysts are less attractive because of their low efficiency and poor stability [22]. Some strategies have been proposed to improve the electrocatalytic activity of nickel oxide for the OER such as the introduction of N dopants into NiO nanosheets [23], embedding nickel/nickel oxide within a N-graphene matrix [24], designing of rich nickel vacancies within mesoporous nickel oxide [25] as well as employing different morphological nanostructures [26,27]. Despite tremendous efforts, the development of efficient and robust OER catalysts based on nickel oxide nanostructures is still a great challenge and needs further investigation. Herein, we introduce a simple post-synthesis approach to prepare highly porous Co-doped NiO nanorods using Co-doped $NiC_2O_4$ nanorods as a precursor. The remarkable features of highly porous NiO nanorods such as large surface area and high porosity may provide better charge transfer, more catalytic active sites on the surface and efficient chemical diffusion. Additionally, the synergic effect of cobalt in the nickel oxide matrix is an essential factor for enhancing catalytic activity for OER.

# 2. Experimental

## 2.1. Synthesis of highly porous undoped and Co-doped NiO nanorods

Nickel(II) chloride ($NiCl_2 \cdot 6H_2O$) and oxalic acid ($H_2C_2O_4 \cdot H_2O$) were used as the starting materials for the preparation of pure $NiC_2O_4$ nanorod precursor and highly porous NiO nanorods. Cobalt(II) nitrate ($Co(NO_3)_2 \cdot 6H_2O$) was used as a dopant precursor for the synthesis of Co-doped $NiC_2O_4$ nanorod precursors and highly porous Co-doped NiO nanorods. The solvent used for the synthesis of these samples was glycerol. All chemicals were purchased from Sigma-Aldrich and used as received without further purification.

Highly porous undoped and Co-doped NiO nanorods were prepared by a simple post-synthesis route. For the synthesis of undoped $NiC_2O_4$ nanorods, 2 mmol $NiCl_2 \cdot 6H_2O$ and 0.7 mmol oxalic acid were dissolved in 9 ml distilled water and 16 ml glycerol by magnetic stirring for 1 h to obtain a clear solution. The solution was transferred to a 50 ml Teflon-lined stainless-steel autoclave, and then aged at 150°C for 12 h. After cooling to room temperature, the solid products were collected by filtration and washed thoroughly by distilled water and ethanol. The precipitates were dried at 80°C for 24 h to obtain $NiC_2O_4$ nanorods. The $NiC_2O_4$ nanorod precursor was annealed at 500°C for 3 h with a heating rate of 2°C min$^{-1}$ to form highly porous NiO nanorods. The Co-doped $NiC_2O_4$ nanorods were synthesized using a similar process as described for pure $NiC_2O_4$. Therein, the concentration of the $Co(NO_3)_2 \cdot 6H_2O$ dopant was 1–7% in molarity as starting sources. The as-synthesized Co-doped $NiC_2O_4$ nanorods were transformed into highly porous Co-doped NiO nanorods by a similar annealing process. The resultant nanomaterials were named as $X$%Co-NiO wherein $X$ represents the amount of Co dopant introduced.

## 2.2. Material characterizations

X-ray diffraction (XRD) measurements of the samples were conducted using a Bruker D8 Advance X-ray diffractometer. The morphology and chemical composition of the products were analysed by scanning electron microscopy (SEM) and energy-dispersive X-ray spectroscopy (EDS) with a JSM-5300LV instrument and high-resolution transmission electron microscopy (HRTEM), transmission electron microscopy (TEM) as well as selected area electron diffraction (SAED) with a JEOL JEM 1230. Thermogravimetric analysis (TG) and differential thermal analysis (DTA) were conducted using a Labsys TG/DTA-SETARAM simultaneous thermogravimetric analyser with a heating rate of 2°C min$^{-1}$ in air environment. A Nicolet 6700 FTIR spectrometer was used to record the infrared spectra of the samples. The Brunauer–Emmett–Teller (BET) specific surface area and Barrett–Joyner–Halenda pore size distribution of the highly porous NiO nanorods were obtained from nitrogen adsorption/desorption isotherm measurement.

The extended X-ray absorption fine-structured (EXAFS) spectra of the Co K-edge (7709 eV) and Ni K-edge (8333 eV) were recorded in fluorescence mode using a Ge (220) double-crystal monochromator at the beam line BL8 at Synchrotron Light Research Institute (SLRI), Nakhon Ratchasima, Thailand (at the 1.2 GeV storage ring, with an average beam current of 80–120 mA) [28]. Measurements were performed at room temperature. All data analysis was performed using ATHENA software.

## 2.3 Catalytic assay

Three milligrams of highly porous NiO nanorod-based catalyst (pure and Co-doped NiO nanorods) was dispersed in a solvent mixture constituted of 0.2 ml ethanol and 0.8 ml deionized water to prepare catalyst ink. It was then drop-cast onto a clean fluorine-doped tin oxide (FTO)-coated glass electrode with NiO nanorod-based catalyst loading of 100 µg cm$^{-2}$. The resultant catalyst-loaded FTO electrode was then naturally dried in air prior to use. For example, a photograph of 3%Co-doped NiO nanorod catalyst-loaded FTO electrode is shown in the electronic supplementary material, figure S1.

The activity of the catalyst electrode was assayed in a pH 14 NaOH solution using a conventional three-electrode configuration with the catalyst working electrode, Ag/AgCl in 1 M KCl as reference electrode and a Pt mesh counter electrode. Linear sweep voltammograms were recorded with a potential scan rate of 2 mV s$^{-1}$. All potentials were quoted against the reversible hydrogen electrode (RHE).

# 3. Results and discussion

## 3.1 Synthesis and characterization of highly porous NiO nanorods with and without Co dopant

As revealed by SEM analysis, the pristine $NiC_2O_4 \cdot 2H_2O$ precursor consists of nanorods with an average diameter of 50–100 nm and length of 0.5–1 µm (electronic supplementary material, figure S2(a)). The respective TEM images indicate that the surface of nanorods is smooth (electronic supplementary material, figure S3(a,b)). Introduction of Co dopant at 1–5 at% did not induce a significant change in the morphology of $NiC_2O_4 \cdot 2H_2O$ nanorods (electronic supplementary material, figure S2(b–f)). However, at higher content of Co, the morphology of the sample was changed significantly. Actually, the 6%Co-$NiC_2O_4$ and 7%Co-$NiC_2O_4$ samples showed irregular shapes (electronic supplementary material, figure S2(g,h)), and the length of rods decreased in comparison with that of 0–5%Co samples. A large number of nanoparticles were observed clearly in the SEM images of 6%Co-$NiC_2O_4$ and 7%Co-$NiC_2O_4$ samples.

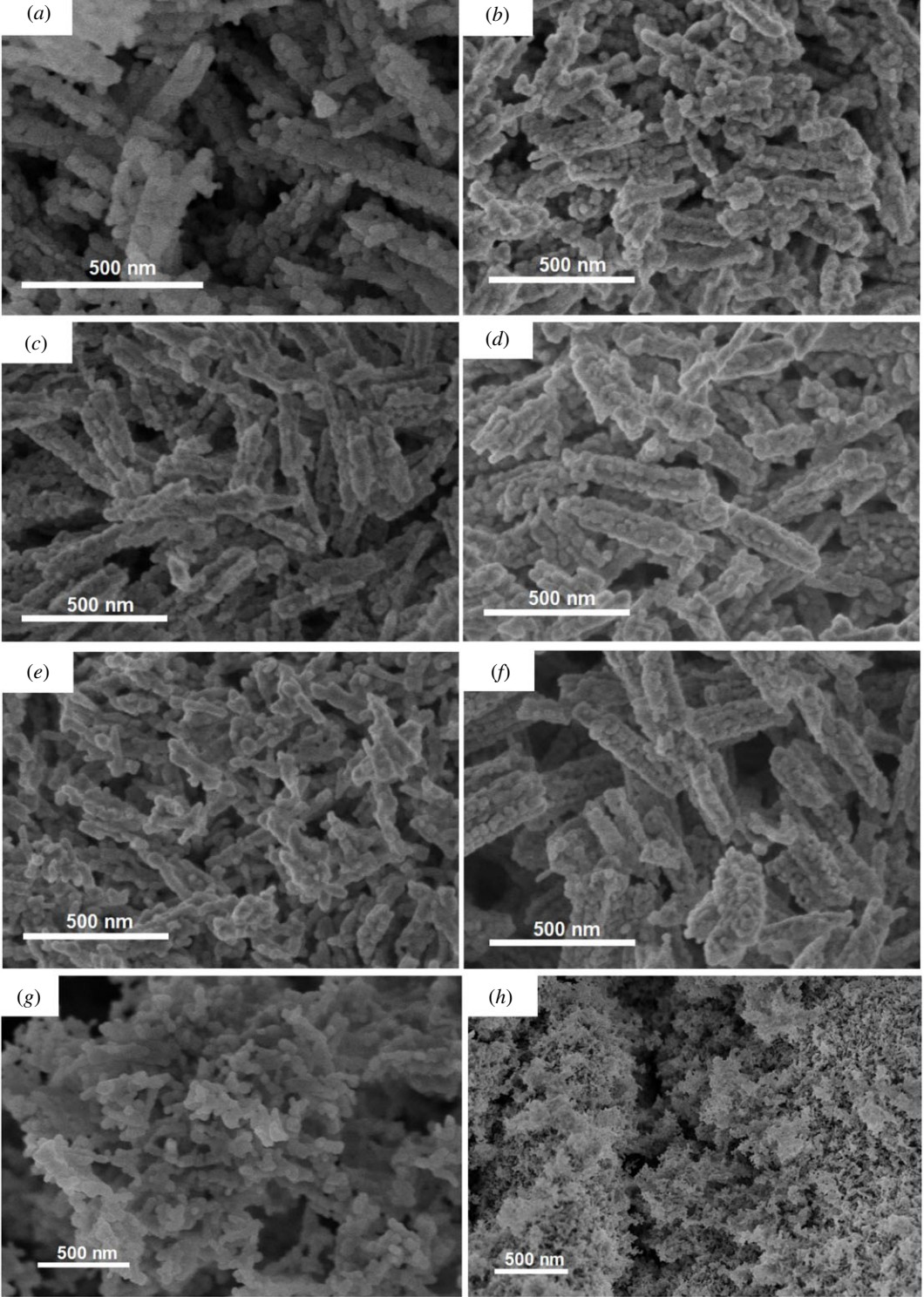

**Figure 1.** SEM images of calcined porous NiO nanorods (*a*) and Co-doped porous NiO nanorod samples: (*b*) 1%Co-NiO, (*c*) 2%Co-NiO, (*d*) 3%Co-NiO, (*e*) 4%Co-NiO, (*f*) 5%Co-NiO, (*g*) 6%Co-NiO and (*h*) 7%Co-NiO.

The calcination resulted in a drastic change of morphology. The calcined products with Co dopant content ranging from 0 to 5 at% consisted of porous rod-like nanostructures with rough surface (figure 1*a–f*). The TEM images of samples (figure 2*a–f*) revealed that the nanorods were constructed of nanosized particles as building blocks. The clear contrast between dark and light regions in an individual rod of pure NiO (electronic supplementary material, figure S3(*c*)) implies the formation of porous nanorods which could be originated from the oxidation reaction of the organic component as well as phase transformation of the precursors into NiO during annealing under air

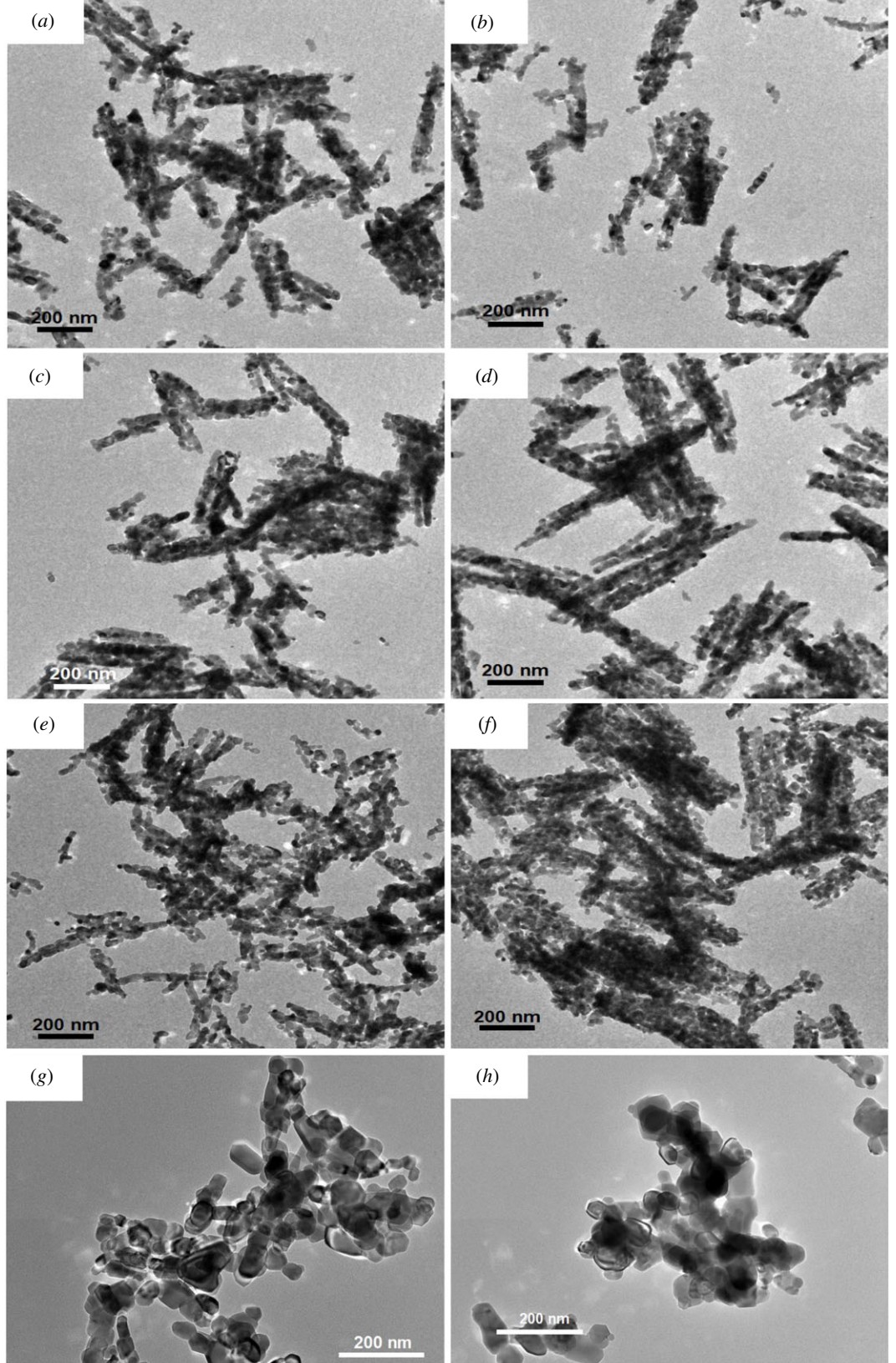

**Figure 2.** TEM images of calcined porous NiO nanorods (*a*) and Co-doped porous NiO nanorod samples: (*b*) 1%Co-NiO, (*c*) 2%Co-NiO, (*d*) 3%Co-NiO, (*e*) 4%Co-NiO, (*f*) 5%Co-NiO, (*g*) 6%Co-NiO and (*h*) 7%Co-NiO.

environment [29]. For 6%Co-NiC$_2$O$_4$ and 7%Co-NiC$_2$O$_4$ samples, the nanorods were broken and transformed into particle agglomerates, as observed in SEM (figure 1*g,h*) and TEM images (figure 2*g,h*). The shape inhomogeneity may be related to the strain induced in the lattice by the increase of

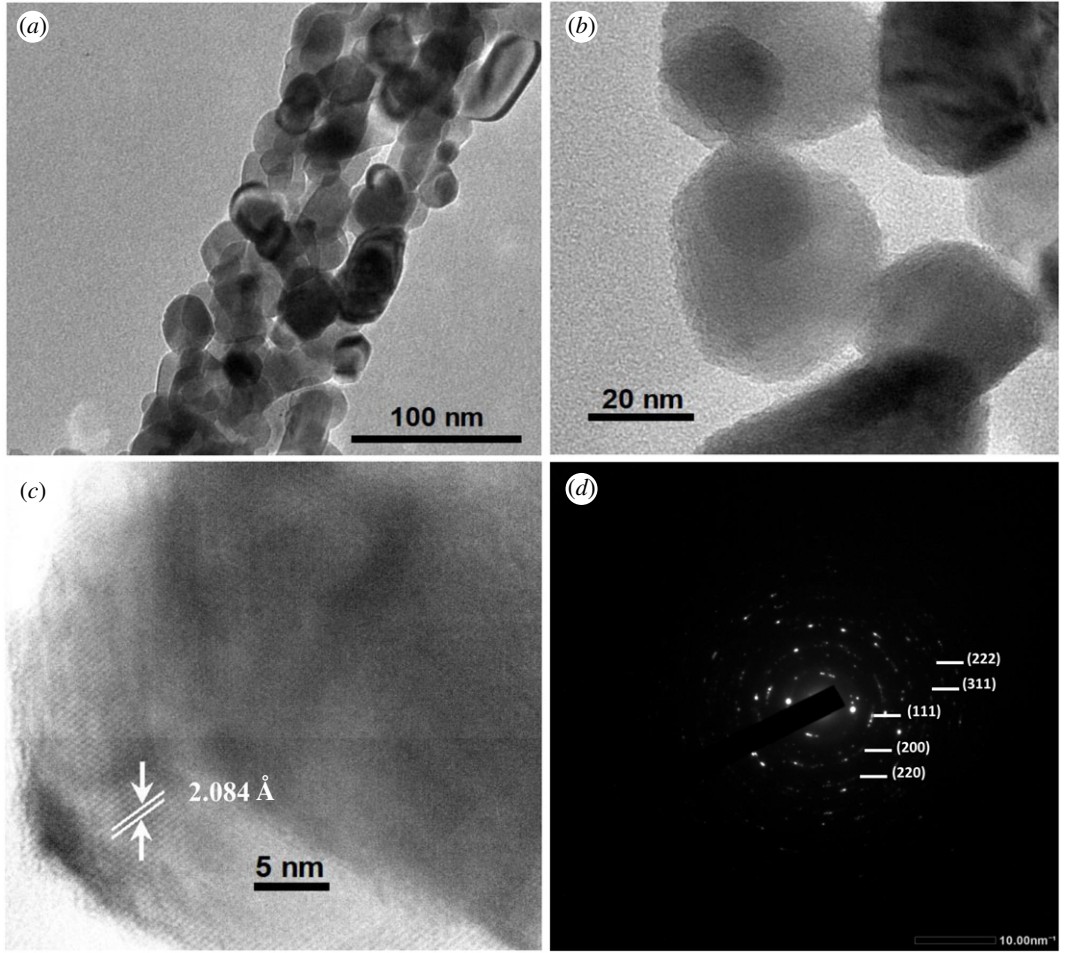

**Figure 3.** TEM image (*a*), HRTEM images (*b,c*) and SAED pattern (*d*) of 5%Co-doped NiO nanorods.

Co concentration or the formation of highly reactive, weak bonding environments when a cobalt cation attaches to a surface site [30].

The TEM image of a single rod of 5%Co-doped NiO nanorod sample (figure 3*a*) shows a similar morphology to that of a pure NiO nanorod sample. The nanopores on the doped nanorods were observed significantly, formed by the aggregation of nanoparticles (figure 3*b*). The HRTEM analysis of 5%Co-doped NiO sample showed its lattice spacing values of 2.084 Å, assigning to the (200) plane face centred cubic NiO, and its SAED pattern corresponding to crystalline nature and phase purity (figure 3*c,d*).

In order to substantiate the formation of porous NiO nanorods under the annealing process, the thermal characterization of the precursor $NiC_2O_4$ nanorods was analysed by TG-DTA. Two weight loss steps were observed clearly in TG-DTA curves (electronic supplementary material, figure S4(a)). The first step with 9.04% weight loss in the temperature range of 30–230°C can be attributed to the evaporation of intercalated water molecules to form anhydrous oxalate. On further increasing the temperature, the second step showing *ca* 44.49% of weight loss in the range of 230–500°C with a sharp exotherm at 324°C in DTA curves was observed. It is assignable to the decomposition reaction of anhydrous nickel oxalate into nickel oxide (equation (3.2)) [31–33]. No other peak was observed beyond 500°C, implying the complete decomposition of precursor $NiC_2O_4 \cdot 2H_2O$ nanorods into porous NiO nanorods. The thermal decomposition process can be explained by the following equations [32]:

$$NiC_2O_4 \cdot 2H_2O \rightarrow NiC_2O_4 + 2H_2O \tag{3.1}$$

and

$$NiC_2O_4 + \frac{1}{2}O_2 \rightarrow NiO + 2CO_2. \tag{3.2}$$

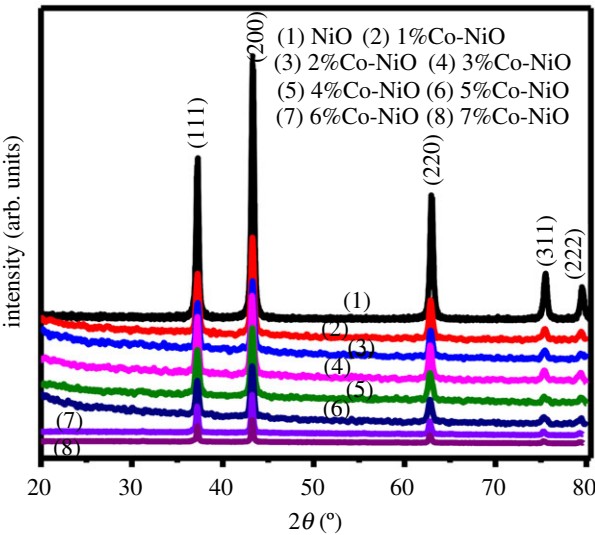

**Figure 4.** XRD patterns of calcined samples 0–7%Co-NiO.

The XRD patterns of precursor samples are shown in the electronic supplementary material, figure S4(b). The XRD pattern of undoped sample shows diffraction peaks at 22.6°, 30.4°, 35.5°, 40.9°, 43.8°, 47.7°, 48.7° and 37.73° that are perfectly indexed to (002), ($\bar{4}$02), (402), (021), ($\bar{3}$14), ($\bar{2}$23), ($\bar{6}$04) and (023) crystal planes of monoclinic nickel oxalate dihydrate (JCPDS card no. 25-0582). In the Co-doped samples, characteristic peaks of cobalt oxalate, cobalt oxide, cobalt hydroxide or other impurities are not visible. The XRD patterns of calcined samples are shown in figure 4. These patterns are well matched to the standard cubic nickel oxide (JCPDS card no. 04-0835), indicating that the $NiC_2O_4$ precursor was completely transformed into NiO phase at 500°C. No obvious peaks of impurities were present in the XRD patterns. Slight shifts in XRD peak position of (200) crystal planes of the cubic NiO phase of Co-doped NiO samples towards lower angle were observed (electronic supplementary material, figure S5), demonstrating the incorporation of Co dopant atoms in the host matrix [34]. The similar radius of Co (0.79 Å) and Ni (0.83 Å) atoms makes Co atoms substitute Ni atoms in lattices of NiO [35], leading to Co ions being highly dispersed in the matrix of NiO.

In the FTIR analysis, the $NiC_2O_4$ and Co-doped $NiC_2O_4$ samples with cobalt dopant range of 1–7% show characteristic bending vibration bands of nickel oxalate dihydrate (electronic supplementary material, figure S6(a,b)). For 0–5%Co-$NiC_2O_4$ samples (electronic supplementary material, figure S5(a)), the absorption band at about 3400 cm$^{-1}$ was assigned to the O–H stretching vibration of hydrate and absorbed water [36]. The peak at approximately 1627 cm$^{-1}$ could be related to both antisymmetric C=O stretching vibration and bending vibration of hydration water [37]. The symmetric C=O stretching vibration band was found at 1357 cm$^{-1}$. In addition, the absorption bands at 1317 cm$^{-1}$ and 829 cm$^{-1}$ were assigned to the C–O and C–C stretching vibrations of coordinated oxalic acid, respectively [38]. The Ni–O bending of oxalate moiety was assigned to the absorption band at about 491 cm$^{-1}$ [39]. The FTIR spectra of 6%Co-$NiC_2O_4$ and 7%Co-$NiC_2O_4$ showed similar features to that of $NiC_2O_4$ (electronic supplementary material, figure S6(b)), excepting slight shifts were observed for the symmetric C=O, C–O, C–C and Ni–O bending vibration bands. After the calcination process, the feature absorption bands of $NiC_2O_4$-based samples disappeared in FTIR spectra of calcined samples (electronic supplementary material, figure S6(c,d)). For 0–5%Co-NiO samples (electronic supplementary material, figure S6(c)), the strong band at about 416 cm$^{-1}$ was assignable to the stretching vibration of Ni–O [40]. The broad absorption band centred at 3400 cm$^{-1}$ and the band at approximately 1627 cm$^{-1}$ were indexed to O–H and H–O–H bending vibration modes [41]. When the Co-dopant concentration was increased, the Ni–O stretching vibration band was slightly shifted (electronic supplementary material, figure S6(d)), which may be related to the defect states of NiO structure [42]. The FTIR spectra further confirmed the successful conversion of undoped and Co-doped $NiC_2O_4 \cdot 2H_2O$ nanorods to undoped and Co-doped porous NiO nanorods, respectively.

In order to demonstrate clearly the incorporation of Co within Co-$NiC_2O_4$ precursors and Co-NiO catalysts, the elemental composition of products was analysed by EDS as shown in the electronic supplementary material, figures S7 and S8. As compared to $NiC_2O_4$, the Co-doped $NiC_2O_4$ samples

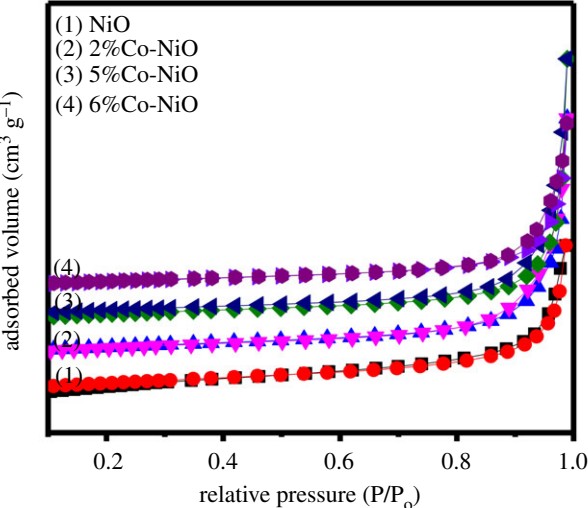

**Figure 5.** N$_2$ adsorption/desorption isotherms of pure highly porous NiO nanorods, and of 2%Co-NiO, 5%Co-NiO and 6%Co-NiO samples.

**Figure 6.** Ni-K edge EXAFS spectra of 5%Co-NiO as-prepared, 5%Co-NiO after catalysis and NiO reference (*a*) and Fourier transform of the experimental EXAFS spectra in K space (*b*); Co-K edge EXAFS spectra of 5%Co-NiO as-prepared, 5%Co-NiO after catalysis and CoO reference (*c*) and Fourier transform of the experimental EXAFS spectra in K space (*d*).

exhibited the presence of a significant amount of Co along with Ni, O and C as a result of cobalt ion incorporation during the synthesis. Cobalt/nickel atomic ratio is very close to the theoretically calculated value in all samples. After the calcining process, the pure NiO confirms the presence of Ni and O elements (electronic supplementary material, figure S9(a)) with a Ni/O atomic ratio of

**Table 1.** Composition of calcined samples measured from EDS.

| samples | | | | | | | | |
|---|---|---|---|---|---|---|---|---|
| | NiO | | 1%Co-NiO | | 2%Co-NiO | | 3%Co-NiO | |
| elements | wt% | at% | wt% | at% | wt% | at% | wt% | at% |
| O K | 23.30 | 52.72 | 23.44 | 52.89 | 23.35 | 52.77 | 23.55 | 53.05 |
| Co K | — | — | 0.83 | 0.51 | 1.43 | 0.88 | 2.31 | 1.41 |
| Ni K | 76.70 | 47.28 | 75.73 | 46.6 | 75.22 | 46.35 | 74.14 | 45.54 |
| total | 100.00 | 100.00 | 100.00 | 100.00 | 100.00 | 100.00 | 100.00 | 100.00 |
| samples | | | | | | | | |
| | 4%Co-NiO | | 5%Co-NiO | | 6%Co-NiO | | 7%Co-NiO | |
| elements | wt% | at% | wt% | at% | wt% | at% | wt% | at% |
| O K | 22.49 | 51.56 | 22.3 | 51.29 | 22.04 | 50.92 | 22.13 | 51.04 |
| Co K | 3.05 | 1.90 | 3.82 | 2.39 | 4.45 | 2.79 | 5.02 | 3.15 |
| Ni K | 74.46 | 46.54 | 73.88 | 46.32 | 73.51 | 46.29 | 72.85 | 45.81 |
| total | 100.00 | 100.00 | 100.00 | 100.00 | 100.00 | 100.00 | 100.00 | 100.00 |

approximately 1 : 1 (table 1). The doped samples showed the existence of Ni, O and Co, indicating the successful doping (electronic supplementary material, figure S9(b–h)). The Co/Ni atomic ratio is basically consistent with the Co-doping concentration in starting experiment (table 1).

The textural features of the 2%Co-NiO, 5%Co-Ni and 6%Co-NiO samples and that of the undoped NiO were explored by nitrogen adsorption/desorption isotherm analysis. Results are presented in figure 5. In all samples, the $N_2$ adsorption/desorption curves, which possess the same shapes, are typical type IV isotherms with an H1 type loop hysteresis, confirming a mesoporous structure in the nanomaterials [43]. The BET surface area of NiO nanorods was $33 \, m^2 \, g^{-1}$, which represented a significant increment in comparison with the $NiC_2O_4 \cdot 2H_2O$ nanorods ($S_{BET}$ approx. $17 \, m^2 \, g^{-1}$) (electronic supplementary material, figure S9). The BET specific surface areas of 2%Co-NiO and 5%Co-NiO rod-like nanostructures were 35 and $32 \, m^2 \, g^{-1}$, respectively, which were slightly changed in comparison with that of pure NiO nanorods. While the specific surface area of 6%Co-NiO sample slightly decreased ($28 \, m^2 \, g^{-1}$), which might be because the highly porous nanorod was broken and agglomerated nanoparticles were formed after the annealing process. The EXAFS spectrum of the Ni-K edge recorded for the 5%Co-NiO nanorod sample (figure 6a, red trace) showed identical features to those of a NiO reference crystal (figure 6a, black trace). The Fourier transform spectral analysis in the $k$ range between 2.5 and 11.5 $\text{Å}^{-1}$ showed a similar Ni–Ni distance between the 5%Co-NiO and the NiO reference sample (figure 6b). However, a small alternation was observed in the shorter range, being assignable to Ni–O interaction. For the 5%Co-NiO sample, the Co-K edge spectrum showed similar features to that of CoO [44,45] (figure 6c,d, red trace). The signal at radial distance of 2.5 Å can be assigned to Co–Ni distance while Co–O bonds are found in shorter distances of 1–2 Å. From the available data, we conclude highly porous NiO nanorods and highly porous Co-doped nanorods were successfully prepared. We then investigated the contribution of the Co dopant to the electrochemical catalytic activity of the NiO nanorods for the OER in alkaline solution.

## 3.2. Catalytic activity of Co-doped highly porous NiO nanorods

The catalytic activity of NiO nanorod-loaded FTO electrode was assayed in an alkaline solution being saturated with oxygen. A NiO nanorod electrode without Co dopant showed a negligible catalytic current density (figure 7a, red trace). The introduction of Co dopant altered the electrochemical property and therefore the catalytic activity of NiO nanorods (figure 7a). The 5%Co-NiO sample shows an oxidation peak at 1.5 V and a shoulder at 1.4 V versus RHE prior to the catalytic event at onset potential of 1.55 V versus RHE, representing an onset overpotential of 320 mV. This onset overpotential is comparable to that determined for a $FeNiO_x$ catalyst [46]. The pre-catalytic oxidation peak can be assigned to $Co^{III}/Co^{II}$ redox couple. This oxidation event shifted as a function of the amount of Co

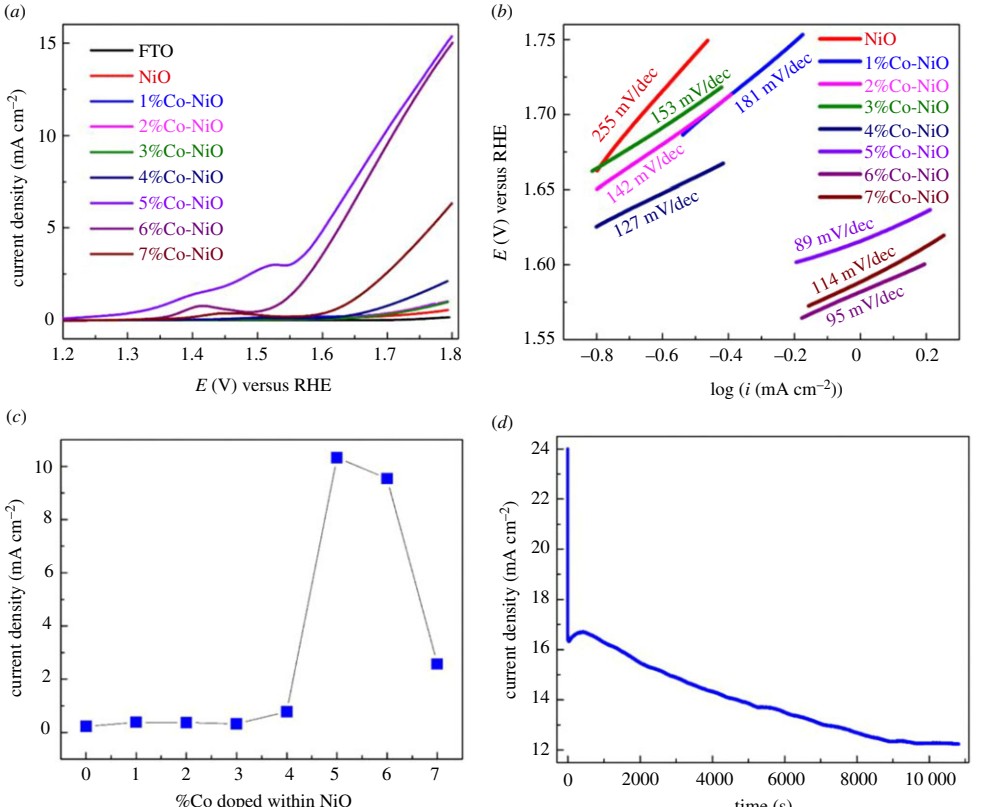

**Figure 7.** $O_2$-evolving catalytic activity of NiO nanorods without and with different Co dopant content assayed in a NaOH electrolyte solution. (a) Linear sweep voltammograms recorded with a potential scan rate of 2 mV s$^{-1}$. (b) Tafel plots. (c) Plot of catalytic current collected at 1.7 V versus RHE as a function of amount of Co dopant. (d) Chronoamperometry recorded at 1.7 V versus RHE.

dopant introduced, likely due to different incorporation within the crystal structure of NiO nanorods. A variation of Co$^{III}$/Co$^{II}$ oxidation peak as a function of Co dopant content has been reported for the case of CoFeOOH catalyst [47]. In the kinetic limiting regime, 5%Co-NiO showed the lowest value of Tafel slope of 89 mV decade$^{-1}$ (figure 7b). To generate a catalytic current density of 10 mA cm$^{-2}$, this catalyst required an overpotential of 450 mV. We then pick up the catalytic current density generated at 1.7 V versus RHE as a descriptor to track the contribution of Co dopant to the catalytic activity. It can be seen that 5% of Co dopant shows the optimal contribution (figure 7c). Given that the 5%Co-NiO sample has a slightly lower surface area than the pristine NiO and 2%Co-NiO samples, this result clearly evidences the role of Co dopant in promoting the catalytic activity of NiO nanorods.

We then investigated the robustness of the 5%Co-doped NiO nanorod catalyst during the catalytic operation. To do so, the catalyst electrode was held at 1.7 V versus RHE (figure 7d) in the electrolyte solution for 3 h. The catalytic current density showed an obvious degradation over time. This could be linked to the physical detachment of catalyst powder from the electrode surface (together with the $O_2$ bubbles generated) or the degradation of the catalyst itself during the catalytic operation. To examine the latter possibility, the morphology and chemical nature of the 5%Co-NiO catalyst after catalysis were examined by SEM and EXAFS. No obvious change in the nanorod morphology was observed as revealed by SEM analysis (electronic supplementary material, figure S10). Furthermore, EDS of all Co-doped NiO nanostructures after catalysis (electronic supplementary material, figure S11) indicated that the cobalt was not washed away during the reaction, confirming that Co was tightly bound in the NiO crystal lattice. EXAFS analysis in the Ni-K and Co-K edges showed similar features for the as-prepared 5%Co-NiO nanorods and the same sample after being operated for 3 h (figure 6a,b, red and blue traces). This suggests intact chemical environments of the Ni and Co within the 5%Co-NiO nanorod catalyst after the catalysis. Only small deviations were observed in the short-range interactions, corresponding to Ni–O and Co–O bonds (figure 6c,d).

In general, the catalytic activity of metal oxide nanostructured electrocatalysts can be improved by controlling the morphologies and doping by metal because they possess an increased active surface area and synergistic interaction [48,49]. The role of Co doping within catalysts for enhanced OER has

been reported for $MoS_2$ [50], $RuO_2$ [51], $TiO_2$ [52] and $SnS_2$ [53], attributed to some effects such as additional catalytic active sites and modifying the electronic structure of catalysts. In this report, the optimal cobalt content added into NiO nanorods was about 5 at% to modify their electronic structures without breaking their nanorod structures. Thus, we speculate that the 5%Co-doped NiO nanorods showed an enhanced activity for OER thanks to the combination of the porous nanoarchitecture and the effect of Co dopant that could provide more abundant catalytic active sites.

## 4. Conclusion

Highly porous Co-doped NiO nanorods with high specific surface area and abundant nanopores were synthesized by a facile post-synthesis route using Co-doped $NiC_2O_4$ nanorods as a precursor. With a reasonable content of Co dopant added into NiO crystal structure (5%Co-NiO sample), the highly porous NiO nanorods are still maintained. The obtained nanocomposite showed a significant increase of catalytic activity for OER, which may be attributed to a combination of highly porous NiO nanoarchitecture and the effect of Co dopant. The result demonstrates the important role of highly porous nanostructures and doping composition of electrocatalysts for enhancing catalytic activity and providing efficient and inexpensive catalysts for OER.

Data accessibility. Our data are available from the Dryad Digital Repository: https://doi.org/10.5061/dryad.g79cnp5p8.
Authors' contributions. N.D.C. provided the research plan, assisted in scientific discussion, performed the experiments, analysed the data and contributed to writing the manuscript. T.D.T performed the experiments, prepared the figures and contributed to writing the manuscript. Q.T.N. helped in performing experiments and assisted in manuscript writing. H.V.M.H and T.T.H. helped in performing the experiments, prepared figures and assisted in the manuscript preparation. D.T.Q and W.K. analysed the data and assisted in scientific discussion. P.D.T. provided the research plan, assisted in scientific discussion and wrote the manuscript. All authors reviewed the manuscript.
Competing interests. We declare we have no competing interests.
Funding. This work was supported by the Vietnam National Foundation for Science and Technology Development (NAFOSTED) under grant no. 103.02-2019.43.
Acknowledgement. We thank the SLRI BL8 staff for their experimental support.

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
