## [Peer Review File · Royal Society Open Science]

Review History

RSOS-202352.R0 (Original submission)

Review form: Reviewer 1 (Dongmei Han)

Is the manuscript scientifically sound in its present form?

No

Are the interpretations and conclusions justified by the results?

No

Is the language acceptable?

Yes

Do you have any ethical concerns with this paper?

Yes

Have you any concerns about statistical analyses in this paper?

No

Recommendation?

Major revision is needed (please make suggestions in comments)

Comments to the Author(s)

The present manuscript prepared porous Co-doped NiO nanorods by hydrothermal route and applied to the oxygen evolution reaction as catalysts. This work can give us some interesting results of preparing nanomaterials for OER catalysts with excellent performance, but I think this manuscript needs some modifications before publication in this journal. Detailed comments are as follows:

1. XRD pattern and FTIR spectra of 0-5%Co-NiC₂O₄ precursor samples may be put into the Supporting Information as secondary information.
2. Please explain and analyze the resistance of the absorption bands at about 1627 and 3400 cm⁻¹ for Co-doped porous NiO samples in the FTIR analysis.
3. Authors state that The catalytic activity of Co-doped NiO could originate from large surface area of Co-doped NiO nanorods, but the BET surface area of Co-doped NiO samples were not provided. It is vital to come to the conclusion with these results.
4. From the catalytic activity of Co-NiO nanorods, O₂ evolving catalytic activity for 1-4% Co-NiO samples are very close, while the activity for 5% Co-NiO samples is much higher than 1-4% Co-NiO. Even the activity for 7% Co-NiO samples is slightly higher than 1-4% Co-NiO. So, the investigation of internal factors should be carried out for the samples of 4-7% Co-NiO.
5. It's a huge difference of the catalytic activity between 4% and 5% Co-NiO samples. What is the main reason?
6. The mechanism discussion should be realized by some real data, other than simple reasoning. For example, the authors attribute the improved performance to the important role of Co dopant that could provide more catalytic active sites, but the supporting result was not provide in the section of "Results and discussions".

Review form: Reviewer 2

Is the manuscript scientifically sound in its present form?

Yes

Are the interpretations and conclusions justified by the results?

Yes

Is the language acceptable?

Yes

Do you have any ethical concerns with this paper?

No

Have you any concerns about statistical analyses in this paper?

No

Recommendation?

Major revision is needed (please make suggestions in comments)

Comments to the Author(s)

Author should provide the results above 5% cobalt doping.
Provide the XPS spectra to all as prepared and calcinated samples

Decision letter (RSOS-202352.R0)

Dear Dr Cuong:

Title: Highly porous Co-doped NiO nanorods: Facile hydrothermal synthesis and electrocatalytic oxygen evolution properties
Manuscript ID: RSOS-202352

The editor assigned to your manuscript has now received comments from reviewers. We would like you to revise your paper in accordance with the referee and Subject Editor suggestions which can be found below (not including confidential reports to the Editor). Please note this decision does not guarantee eventual acceptance.

Please submit your revised paper before 28-Jul-2021. Please note that the revision deadline will expire at 00.00am on this date. If we do not hear from you within this time then it will be assumed that the paper has been withdrawn. In exceptional circumstances, extensions may be possible if agreed with the Editorial Office in advance. We do not allow multiple rounds of revision so we urge you to make every effort to fully address all of the comments at this stage. If deemed necessary by the Editors, your manuscript will be sent back to one or more of the original reviewers for assessment. If the original reviewers are not available we may invite new reviewers.

On behalf of the Subject Editor Professor Anthony Stace and the Associate Editor Dr Annette Trunschke.

RSC Associate Editor:

Comments to the Author:

The work is of potential interest but requires major revisions. The surface area of the entire series needs to be measured to support the major conclusion.

The authors observed degradation during OER, but only one spent catalyst was analysed. All spent catalysts should be analysed at least in terms of chemical composition to see if there are differences in leaching. Apparently, doping with Co is important, as maximum performance was observed at intermediate Co content. Unfortunately, the nature of the Co was not analysed. It would be important to check whether there are already differences in the precursors with regard to the structure of the cobalt or the Co-Ni interaction.

RSC Subject Editor:

Comments to the Author:

(There are no comments.)

Reviewers' Comments to Author:

Reviewer: 1

Comments to the Author(s)

The present manuscript prepared porous Co-doped NiO nanorods by hydrothermal route and applied to the oxygen evolution reaction as catalysts. This work can give us some interesting results of preparing nanomaterials for OER catalysts with excellent performance, but I think this manuscript needs some modifications before publication in this journal. Detailed comments are as follows:

1. XRD pattern and FTIR spectra of 0-5% Co-NiC₂O₄ precursor samples may be put into the Supporting Information as secondary information.
2. Please explain and analyze the resistance of the absorption bands at about 1627 and 3400 cm⁻¹ for Co-doped porous NiO samples in the FTIR analysis.
3. Authors state that The catalytic activity of Co-doped NiO could originate from large surface area of Co-doped NiO nanorods, but the BET surface area of Co-doped NiO samples were not provided. It is vital to come to the conclusion with these results.
4. From the catalytic activity of Co-NiO nanorods, O₂ evolving catalytic activity for 1-4% Co-NiO samples are very close, while the activity for 5% Co-NiO samples is much higher than 1-4% Co-NiO. Even the activity for 7% Co-NiO samples is slightly higher than 1-4% Co-NiO. So, the investigation of internal factors should be carried out for the samples of 4-7% Co-NiO.
5. It's a huge difference of the catalytic activity between 4% and 5% Co-NiO samples. What is the main reason?
6. The mechanism discussion should be realized by some real data, other than simple reasoning. For example, the authors attribute the improved performance to the important role of Co dopant that could provide more catalytic active sites, but the supporting result was not provide in the section of "Results and discussions".

Reviewer: 2

Comments to the Author(s)

Author should provide the results above 5% cobalt doping.

Provide the XPS spectra to all as prepared and calcinated samples

Author's Response to Decision Letter for (RSOS-202352.R0)

See Appendix A.

Decision letter (RSOS-202352.R1)

Dear Dr Cuong:

Title: Highly porous Co-doped NiO nanorods: Facile hydrothermal synthesis and electrocatalytic oxygen evolution properties
Manuscript ID: RSOS-202352.R1

It is a pleasure to accept your manuscript in its current form for publication in Royal Society Open Science. The chemistry content of Royal Society Open Science is published in collaboration with the Royal Society of Chemistry.

Yours sincerely,
Dr Ellis Wilde
Publishing Editor, Journals

On behalf of the Subject Editor Professor Anthony Stace and the Associate Editor Dr Annette Trunschke.

RSC Associate Editor
Comments to the Author:
(There are no comments.)

Reviewer(s)' Comments to Author:

Appendix A

University of Sciences, Hue University
77 Nguyen Hue Street, Hue, Vietnam

Dear Prof. Laura Smith,

Thank you for your response to our manuscript to, Royal Society Open Science (ID: RSOS-202352), entitled “Highly porous Co-doped NiO nanorods: Facile hydrothermal synthesis and electrocatalytic oxygen evolution properties”. We sincerely appreciate you remaining the opportunity to consider our paper for publication. RSC Associate Editor recommended that the work is of potential interest. The first reviewer recommended to accept our manuscript after some modifications. He/she liked our work and also evaluated the manuscript possesses some interesting results of preparing nanomaterials for OER catalysts with excellent performance. The second reviewer gave constructive comments to further strengthen the manuscript. Below is our careful revision in response to these comments, which is also involved in the present manuscript for uploading. Any modification in the manuscript was highlighted as **blue color**.

I hope that we have satisfactorily addressed the comments and the revised manuscript is suitable for publication in Royal Society Open Science. I look forward to hearing from you.

Dr. Nguyen Duc Cuong

RSC Associate Editor:

Comments to the Author:

The work is of potential interest but requires major revisions. The surface area of the entire series needs to be measured to support the major conclusion. The authors observed degradation during OER, but only one spent catalyst was analysed. All spent catalysts should be analysed at least in terms of chemical composition to see if there are differences in leaching. Apparently, doping with Co is important, as maximum performance was observed at intermediate Co content. Unfortunately, the nature of the Co was not analysed. It would be important to check whether there are already differences in the precursors with regard to the structure of the cobalt or the Co-Ni interaction.

<Response> We would like to thank Editor for your helpful comments. Based these comments, we have supplied data that include EDX results of Co-doped NiC₂O₄ nanorod precursors (figure S7), EDX results of all samples after catalysis (figure S11) and nitrogen adsorption/desorption isotherm of 2%Co-NiO, 5-6%Co-NiO nanorod samples (figure 5) in the revised version. We believe that our revised version with further supported data is of highest quality for publication in Royal Society Open Science. Please see the revised version.

Reviewer: 1

Comments to the Author (s)

The present manuscript prepared porous Co-doped NiO nanorods by hydrothermal route and applied to the oxygen evolution reaction as catalysts. This work can give us some interesting results of preparing nanomaterials for OER catalysts with excellent performance, but I think this manuscript needs some modifications before publication in this journal. Detailed comments are as follows:

<Response> First, we would like to thank the reviewer for his/her evaluation of our manuscript and giving some valuable comment. Per the reviewer's comments, we have carefully revised our manuscript. Herein, for easily an addressable, we responded the reviewer's comments as question and answer as shown below. We hope that the new version will be acceptable for publication in Royal Society Open Science.

1. XRD pattern and FTIR spectra of 0-5%Co-NiC₂O₄ precursor samples may be put into the Supporting Information as secondary information.

<Response> The suggested correction has been implemented in the revised manuscript.

2. Please explain and analyze the resistance of the absorption bands at about 1627 and 3400 cm⁻¹ for Co-doped porous NiO samples in the FTIR analysis.

<Response> We thank the reviewer for pointing this out. The broad absorption band centered at 3400 cm⁻¹ and the band at ~1627 cm⁻¹ is related to O-H and H-O-H bending vibrations mode (DOI: 10.1016/j.matchemphys.2007.11.031), which index the presence of traces of water in the sample due to absorbed moisture. The results have been explained in the revised version as following:

The broad absorption band centered at 3400 cm⁻¹ and the band at ~1627 cm⁻¹ is indexed to O-H and H-O-H bending vibrations mode.⁴¹

3. Authors state that the catalytic activity of Co-doped NiO could originate from large surface area of Co-doped NiO nanorods, but the BET surface area of Co-doped NiO samples were not provided. It is vital to come to the conclusion with these results.

<Response> We would like to thank you for your helpful comment. Per nice comment, we have added the BET surface area of the samples of 2%Co-NiO and 5-6%Co-NiO to discuss clearly the remarkable electrocatalytic performance of 5%Co-NiO nanorods for OER. Some sentences were added in the revised version as following:

The textural feature of the 2%Co-NiO, 5%Co-Ni and 6%Co-NiO samples and that of the pristine NiO were carried out by nitrogen adsorption/desorption isotherm and results presented in figure 5. In all samples, the N₂ adsorption-desorption curves, which possesses the same shapes, exhibit a typical IV isotherm with an H1 type loop hysteresis, confirming a mesoporous structure in the nanomaterials.⁴³ The BET surface area of NiO nanorods was 33 m²g⁻¹, which represented a significant increment in comparison with the NiC₂O₄·2H₂O nanorods (S_{BET}~ 17 m²g⁻¹) (figure S9). The BET specific surface area of 2%Co-NiO and 5%Co-NiO rod-like nanostructures is respective 35 and 32 m²g⁻¹, which slightly changes in comparison with that of pure NiO nanorods. While the specific surface area of 6%Co-NiO sample slightly decreases (28 m²g⁻¹),

which might be because the highly porous nanorod is broken and agglomerated nanoparticles are formed after annealing process.

4. From the catalytic activity of Co-NiO nanorods, O₂ evolving catalytic activity for 1-4% Co-NiO samples are very close, while the activity for 5% Co-NiO samples is much higher than 1-4% Co-NiO. Even the activity for 7% Co-NiO samples is slightly higher than 1-4% Co-NiO. So, the investigation of internal factors should be carried out for the samples of 4-7% Co-NiO.

5. It's a huge difference of the catalytic activity between 4% and 5% Co-NiO samples. What is the main reason?

<Response> I would like to thank you for nice comments. Because the comments no. 4 and no. 5 are related together. We would like to combine these two comments to answer. In generally, the doping process of suitable element into electrocatalysts can modulate their electronic structure and kinetics of electron transfer at interface, which lead to a promoting their electrocatalytic OER activity (DOI: 10.1039/D0TA07163C). Therefore, adding suitable amount of dopants into electrocatalyst will significantly increase their OER catalytic performance (DOI: 10.1002/ange.201609080; 10.1016/j.mcat.2020.110894) without causing structural collapse. In this report, the optimal cobalt content is added about of 5-6% to modify the electronic structure of NiO nanorod for improving its OER activity. However, when the added cobalt content exceeds the optimal amount, their nanorod structures are broken. This means that the carrier concentration, mobility and distribution of electric charge of NiO nanorod electrocatalyst at interface is changed significantly as well as the specific surface area of catalyst also decrease, leading to a decrease of OER performance of Co-doped NiO catalyst. Per these nice comment, we have added the sentences in the revised version.

In this report, the optimal cobalt content is added into NiO nanorods about of 5% at. to modify their electronic structure without breaking their nanorod structures. Thus, we speculate that the 5%Co-doped NiO nanorods showed an enhanced activity for OER thanks to the combination of the porous nano-architecture and the effect of Co dopant that could provide richer catalytic active sites.

6. The mechanism discussion should be realized by some real data, other than simple reasoning.

For example, the authors attribute the improved performance to the important role of Co dopant that could provide more catalytic active sites, but the supporting result was not provide in the section of “Results and discussions”.

<Response> I would like to thank you for your nice comment. Based your comments and RSC Associate Editor’ comment, we have further suppld data that include EDX results of Co-doped NiC₂O₄ nanorod precursors (figure S7), EDX results of all spent catalyst (figure S11) and specific surface area of 2%Co-NiO and 5-6%Co-NiO nanorods (figure 5) in the revised version to support for the two main factors that enhance the electrocatalytic activity for OER of porous NiO nanorod-based catalyst including: (i) their unique highly porous nanorod, and (ii) the important role of Co dopant. Please see the revised version.

Reviewer: 2

Comments to the Author(s)

First, we would like to thank the reviewer for his/her evaluation of our manuscript and giving some valuable comments. Per these suggestions, we have carefully checked and revised our manuscript. Herein, for easily an addressable, we responded the reviewer’s comments as question and answer as shown below. We hope that the new version will be accepted for publication in Royal Society Open Science.

Author should provide the results above 5% cobalt doping.

<Response> I would like to thank you for your helpful comment. Per this suggestion, we have combined the characteristic results of 6%Co-NiO and 7%Co-NiO samples in figure 1, figure 2, figure 4, and figures S2, S4 (b) S5, S6, S7, S8 and S11 (Please see the revised version)

Provide the XPS spectra to all as prepared and calcinated samples

<Response> Again, we thank the reviewer for pointing this out. We agree with the reviewer about possible contributions of XPS analysis in this study and originally we had planned to carry out XPS analysis of the calcined samples for certify the impact of Co doping in the valence stage of NiO nanorod catalyst. However, in Vietnam, there is no XPS instrument. We often have to send our samples to research groups of foreign countries to measure this characterization. But due to the impact of SARS-CoV-2, it is very difficult for us to send the samples abroad. So, we are not able to get the data even during revised submission. We request the reviewer to kindly consider our helplessness in these unavoidable circumstances.